# Effects of Rare Phytocannabinoids on the Endocannabinoid System of Human Keratinocytes

**DOI:** 10.3390/ijms23105430

**Published:** 2022-05-12

**Authors:** Camilla Di Meo, Daniel Tortolani, Sara Standoli, Clotilde Beatrice Angelucci, Federico Fanti, Alessandro Leuti, Manuel Sergi, Salam Kadhim, Eric Hsu, Cinzia Rapino, Mauro Maccarrone

**Affiliations:** 1Faculty of Bioscience and Technology for Food Agriculture and Environment, University of Teramo, 64100 Teramo, Italy; cdimeo@unite.it (C.D.M.); sstandoli@unite.it (S.S.); ffanti@unite.it (F.F.); msergi@unite.it (M.S.); 2European Center for Brain Research (CERC)/Santa Lucia Foundation IRCCS, 00143 Rome, Italy; daniel.tortolani88@gmail.com (D.T.); a.leuti@unicampus.it (A.L.); 3Faculty of Veterinary Medicine, University of Teramo, 64100 Teramo, Italy; bcangelucci@unite.it; 4Department of Medicine, Campus Bio-Medico University of Rome, 00128 Rome, Italy; 5InMed Pharmaceuticals Inc., Vancouver, BC V6C 1B4, Canada; skadhim@inmedpharma.com (S.K.); ehsu@inmedpharma.com (E.H.); 6Department of Biotechnological and Applied Clinical Sciences, University of L’Aquila, 67100 L’Aquila, Italy

**Keywords:** endocannabinoids, enzymes, keratinocytes, phytocannabinoids, receptors, skin

## Abstract

The decriminalization and legalization of *cannabis* has paved the way for investigations into the potential of the use of phytocannabinoids (pCBs) as natural therapeutics for the treatment of human diseases. This growing interest has recently focused on rare (less abundant) pCBs that are non-psychotropic compounds, such as cannabigerol (CBG), cannabichromene (CBC), Δ^9^-tetrahydrocannabivarin (THCV) and cannabigerolic acid (CBGA). Notably, pCBs can act via the endocannabinoid system (ECS), which is involved in the regulation of key pathophysiological processes, and also in the skin. In this study, we used human keratinocytes (HaCaT cells) as an in vitro model that expresses all major ECS elements in order to systematically investigate the effects of CBG, CBC, THCV and CBGA. To this end, we analyzed the gene and protein expression of ECS components (receptors: CB_1_, CB_2_, GPR55, TRPV1 and PPARα/γ/δ; enzymes: NAPE-PLD, FAAH, DAGLα/β and MAGL) using qRT-PCR and Western blotting, along with assessments of their functionality using radioligand binding and activity assays. In addition, we quantified the content of endocannabinoid(-like) compounds (AEA, 2-AG, PEA, etc.) using UHPLC-MS/MS. Our results demonstrated that rare pCBs modulate the gene and protein expression of distinct ECS elements differently, as well as the content of endocannabinoid(-like) compounds. Notably, they all increased CB_1/2_ binding, TRPV1 channel stimulation and FAAH and MAGL catalytic activity. These unprecedented observations should be considered when exploring the therapeutic potential of cannabis extracts for the treatment of human skin diseases.

## 1. Introduction

Cannabis (*Cannabis sativa*) is a weed that is native to Central and South Asia [1]. It contains more than 500 bioactive components, specifically phytocannabinoids (pCBs) but also polyphenols, flavonoids, terpenes, terpenoids and fatty acids [2,3,4]. The recreational and medicinal properties of cannabis extracts have been known for centuries [5] and today, these extracts are consumed by more than 4% of the global population [6].

Medical cannabis (or medicinal marijuana) has been legalized in many countries, under controlled and regulated terms [7], yet controversial issues remain about its therapeutic efficacy [5]. Moreover, recreational cannabis remains illegal in most countries because of the abusive usage of the primary psychoactive cannabinoid Δ^9^-tetrahydrocannabinol (THC) [7]. The pCBs are prenylated terpenophenolic compounds that are derived from isoprenoid and fatty acid precursors [3,8,9]. In particular, THC and cannabidiol (CBD) are the most abundant and the most extensively studied pCBs [10,11,12]. Cannabichromene (CBC), Δ^9^-tetrahydrocannabivarin (THCV) and cannabigerolic acid (CBGA) are produced along their biosynthetic pathway and part of the latter compound is decarboxylated to cannabigerol (CBG) [13,14]. Growing interest in this topic has recently focused on these rare (less abundant) pCBs; for instance, cannabidiolic acid (CBDA) and CBC are used in creams, foods and beverages [15] and cannabinol has shown neuroprotective effects on an experimental model of glaucoma [16]. Notably, pCBs can act also through the complex endocannabinoid system (ECS) [10,11,17], i.e., the cell-signaling network that is composed of endogenous lipids (endocannabinoids, eCBs), their receptors and metabolic enzymes [9]. These eCBs are involved in the pathophysiological modulation of many central and peripheral functions [10,11,18]. The two most commonly studied eCBs, *N*-arachidonoylethanolamine (anandamide; AEA) and 2-arachidonoylglycerol (2-AG), as well as the eCB-like compound *N*-palmitoylethanolamine (PEA), originate from the degradation of membrane phospholipids and are involved in the regulation of the central and peripheral nervous system, immune responses, food intake, bone homeostasis, reproductive functions and skin biology [18,19]. The eCBs preferably bind to G protein-coupled receptors, such as type-1 (CB_1_) and type-2 (CB_2_) cannabinoid receptors, as well as to the orphan G protein-coupled receptor 55 (GPR55) [20], the transient receptor potential vanilloid type 1 (TRPV1) channels [21] and the peroxisome proliferator-activated receptors α, γ and δ (PPAR α/γ/δ) [22,23]. Moreover, several enzymes are involved in eCB metabolism: AEA is synthesized mainly by *N*-acyl-phosphatidylethanolamines-specific phospholipase D (NAPE-PLD) [24] and is degraded by fatty acid amide hydrolase (FAAH) [25]. Additionally, 2-AG is mainly synthesized by *sn*-1-specific diacylglycerol lipases α and β (DAGL α/β) and is degraded by a specific monoacylglycerol lipase (MAGL) [26,27]. Although our current understanding of the pharmacology and biological activity of rare pCBs remains poor, recent evidence has suggested that they can modulate eCB-binding receptors [28], which could lead to potential benefits in the treatment of skin disorders [17,29,30,31,32,33]. In line with this, different skin-specific cells, such as keratinocytes, melanocytes and sebocytes, represent a relevant target for pCBs [30,34,35,36] and human keratinocytes express all major ECS elements [30,34,37,38].

Against this background, herein we sought to systematically investigate the possible effects of four rare pCBs (CBG, CBC, THCV and CBGA), as shown in Figure 1, on the major ECS elements of human HaCaT cells, which are a widely used model of immortalized keratinocytes [30,34,38]. To this end, the effects of non-cytotoxic doses of these pCBs were ascertained at the gene and protein expression level, as well as in the functional activity of eCB-binding receptors (CB_1_, CB_2_, GPR55, TRPV1 and PPARα/γ/δ) and metabolic enzymes (NAPE-PLD, FAAH, DAGLα/β and MAGL). Moreover, the endogenous content of AEA, 2-AG and PEA was measured in the HaCaT cells that were treated with pCBs under the same experimental conditions.

## 2. Results

### 2.1. Effects of pCBs on Viability, Apoptosis and Cell Cycle of Human HaCaT Cells

Firstly, the effects of CBG, CBC, THCV and CBGA on the viability of HaCaT cells were investigated after treatment at different concentrations (up to 25 μM) and at different time points (up to 24 h) using an MTT cytotoxicity assay (Appendix A). From these experiments, the half maximal inhibitory concentration (IC_50_) of each pCB could be calculated at the different time points, as summarized in Table 1. All pCBs were found to exhibit toxic effects on HaCaT cells at concentrations of >10 µM at all incubation timeframes, with THCV (and even more so CBGA) being the least toxic compound (Table 1). Subsequently, the longest incubation time (24 h) was used to examine the possible effects of pCBs on apoptosis and cell cycles, since eCB signaling is known to play a key role in the proliferation and differentiation of keratinocytes [37]. It was found that CBG and CBGA, at the highest concentration of 25 µM, induced a significant increase in DNA fragmentation (a hallmark of apoptosis). CBG also induced the same effect at 10 µM, whereas THCV was ineffective at all doses (Appendix A). In addition, FACS analysis showed that none of the pCBs affected cell cycle distribution (G0/1, S, G2/M) when used at concentrations of up to 10 μM for 24 h (Appendix A). Incidentally, in the cell cycle assays, we excluded the 25 µM concentration from testing due to its high cytotoxicity, which was observed during the MTT test (Appendix A).

Based on the first set of results, we chose the non-cytotoxic dose of half of the calculated IC_50_ at 24 h to perform all subsequent analyses. These half-IC_50_ values were: 6.0 μM for CBG, 4.0 μM for CBC, 9.0 μM for THCV and 13.0 μM for CBGA.

### 2.2. Effects of pCBs on Gene Expression of ECS Elements in HaCaT Cells

The mRNA expression of the main eCB-binding receptors (CB_1_, CB_2_, GPR55, TRPV1 and PPARα/γ/δ) and the main AEA (NAPE-PLD and FAAH) and 2-AG (DAGLα/β and MAGL) metabolic enzymes was evaluated in the HaCaT cells following treatment with the half-IC_50_ doses of pCBs for 24 h using quantitative real-time reverse transcriptase-polymerase chain reaction (qRT-PCR). The effects of the pCBs on the different receptors and enzymes are shown in Figure 2 and Figure 3, respectively. Overall, CBG was found to significantly upregulate the gene expression of CB_1_ (*p* < 0.0001 vs. control) and CB_2_ (*p* < 0.05 vs. control) whereas the expression of those genes was significantly downregulated by CBC (*p* < 0.0001 vs. control for CB_1_; *p* < 0.01 vs. control for CB_2_). The latter pCB also reduced the gene expression of GPR55 and PPARγ (*p* < 0.0001 and *p* < 0.01 vs. control, respectively). The PPARγ receptor was also significantly reduced by THCV (*p* < 0.0001 vs. control). Moreover, a significant increase in GPR55 gene expression was apparent following the THCV and CBGA treatment (*p* < 0.0001 and *p* < 0.001 vs. control, respectively) (Figure 2a–d).

As far as eCB metabolism was concerned, CBG was found to significantly increase the gene expression of MAGL (*p* < 0.05 vs. control), CBC significantly increased the gene expression of NAPE-PLD (*p* < 0.05 vs. control) and CBGA significantly reduced the gene expression of FAAH (*p* < 0.001 vs. control). THCV and CBGA had no effects on the AEA and 2-AG metabolic enzymes (Figure 3a–d). Overall, the effects of the pCBs on the mRNA levels of metabolic enzymes appeared to be more selective than those on the mRNA levels of receptors, which seems noteworthy when taking into consideration that eCB signaling is subjected to tight metabolic control [9].

### 2.3. Effects of pCBs on Protein Expression of ECS Elements in HaCaT Cells

In addition, the protein expression of the main eCB-binding receptors and metabolic enzymes was measured in the HaCaT cells after treatment with the half-IC_50_ doses of the pCBs for 24 h by means of a densitometric analysis of Western blots. All pCBs significantly increased the protein expression of TRPV1 (*p* < 0.0001 vs. control), while reducing that of CB_2_ (*p* < 0.05 vs. control for CBG and CBC; *p* < 0.001 vs. control for THC0056; *p* < 0.0001 vs. control for CBGA). PPARα/δ, but not PPARγ, protein expression was also significantly reduced (PPARα: *p* < 0.01 vs. control for CBC and THCV; *p* < 0.0001 vs. control for CBGA; PPARδ: *p* < 0.001 vs. control for CBG; *p* < 0.0001 vs. control for CBC, THCV and CBGA). In addition, CB_1_ protein expression was significantly reduced by THCV (*p* < 0.05 vs. control) and CBGA (*p* < 0.0001 vs. control), but not by the other pCBs that were tested (Figure 4).

Moreover, all pCBs significantly increased the expression of NAPE-PLD (*p* < 0.01 vs. control for CBC; *p* < 0.0001 vs. control for CBG, THCV and CBGA). THCV and CBGA also increased the protein expression of MAGL (*p* < 0.0001 vs. control) and DAGLα (*p* < 0.001 vs. control for THCV; *p* < 0.0001 vs. control for CBGA), whereas THCV significantly reduced the expression of DAGLβ (*p* < 0.001 vs. control). Finally, CBC and CBGA significantly increased FAAH protein expression (*p* < 0.01 and *p* < 0.0001 vs. control, respectively) (Figure 5). Incidentally, our protein expression data for the DAGL isozymes were consistent with the notion that non-neuronal cells (such as keratinocytes) predominantly express DAGLβ, whereas DAGLα is more abundant in the brain [39].

The most significant effects of pCBs on the gene and protein expression of ECS elements in HaCaT cells are summarized in Table 2.

### 2.4. Effects of pCBs on Functional Activity of ECS Elements in HaCaT Cells

We also sought to investigate the effects of half-IC_50_ doses of pCBs on the functional activity of eCB-binding receptors (CB_1/2_ and TRPV1) and the AEA and 2-AG metabolizing enzymes (NAPE-PLD, FAAH, DAGLα/β and MAGL) when used to treat HaCaT cells for 24 h, for which an assay procedure was available. To this end, a CB_1/2_ ligand binding assay was performed using the radiolabeled synthetic cannabinoid [^3^H]CP 55,940, which acts as a full agonist of both cannabinoid receptors with high and roughly equal affinity [40].

In these assays, all pCBs significantly increased total CB_1/2_ binding activity compared to the vehicle-treated (control) HaCaT cells (*p* < 0.05 vs. control for CBG, THCV and CBGA; *p* < 0.01 vs. control for CBC), as summarized in Table 3.

In addition, all pCBs significantly increased TRPV1 channel stimulation (*p* < 0.05 vs. control for CBC and CBG; *p* < 0.01 vs. control for THCV and CBGA) (Figure 6), which was measured as intracellular Ca^2+^ release that was triggered by the selective TRPV1 agonist capsaicin [41]. Incidentally, this increased stimulation paralleled a similar increase in the protein expression of TRPV1, which suggested the possibility that it was entirely due to a more abundant (rather than a more active) receptor.

Moreover, only CBGA significantly increased the activity of NAPE-PLD (*p* < 0.0001 vs. control) (Figure 7a) and all pCBs significantly increased the activity of FAAH (*p* < 0.05 vs. control for CBG; *p* < 0.0001 vs. control for CBC, THCV and CBGA) (Figure 7b). CBG, CBC and THCV increased DAGLα/β activity remarkably (*p* < 0.0001 vs. control) (Figure 7c) and all pCBs significantly increased the activity of MAGL (*p* < 0.01 vs. control for CBC; *p* < 0.001 vs. control for CBG; *p* < 0.0001 vs. control for THCV and CBGA) (Figure 7d).

Incidentally, the activity ratios of NAPE-PLD to FAAH and of DAGLα/β to MAGL showed no effects (for CBG) nor a trend toward opposite effects (for CBC, THCV and CBGA), as reported in Table 4.

### 2.5. Endogenous Content of eCBs in HaCaT Cells Treated with pCBs

Finally, the endogenous levels of the major eCBs, AEA and 2-AG, and the major eCB-like compound PEA [11] were measured in the HaCaT cells following treatment for 24 h with the half-IC_50_ doses of pCBs by means of liquid chromatography–mass spectrometry (UHPLC-MS/MS). It was found that only CBG markedly increased 2-AG and PEA contents (*p* < 0.01 and *p* < 0.05 vs. control, respectively), with the other pCBs being ineffective (Figure 8).

## 3. Discussion

This study represents the first systematic analysis of the effects of four rare (less abundant) non-psychotropic pCBs (CBG, CBC, THCV and CBGA) on the major ECS elements, which were investigated in a widely used in vitro model of human keratinocytes (HaCaT cells). These cells express CB_1_, CB_2_, GPR55, TRPV1 and PPARα/γ/δ receptors, as well as NAPE-PLD, FAAH, DAGLα/β and MAGL enzymes. The pCBs are lipophilic substances and hence, they are readily absorbed through the skin, where ECS plays a critical role in controlling epidermal differentiation and cutaneous inflammation [37,42,43,44]. Unsurprisingly, some pCBs are promising drug candidates for the treatment of atopic dermatitis, psoriasis, scleroderma and acne, as well as keratin diseases, skin tumors and pruritus [17,29,32,36]. Furthermore, in recent years, the interest in rare pCBs has increased due to methodological advancements in the extraction/isolation, semi-synthesis or complete synthesis and microbial engineering (in *E. coli*, algae, yeast, etc.) of these compounds. In general, the large-scale production of pCBs has facilitated their use as skin care products [28]. In accordance with this, it is known that pCBs, such as CBD and CBDV, are able to alter in vitro mitochondrial metabolic rates and DNA syntheses at μM concentrations in both cancerous and normal human cells [45]. Moreover, Δ^9^-THC, CBD and CBG have been shown to inhibit the proliferation of transformed human keratinocytes, which are notably different from HaCaT cells in that are under hyperproliferative conditions [46]. Previously, our group showed that CBD and CBG, at μM concentrations, act as transcriptional repressors in the skin, where they control cell proliferation and differentiation by modulating key proteins (keratins 1 and 10, involucrin and transglutaminase 5) through the DNA methylation of their genes [38]. Overall, the available data in the literature have demonstrated that pCBs do indeed have a biological activity in skin cells; nevertheless, the underlying molecular mechanisms remain largely unknown. Herein, we assessed cytotoxicity of CBG, CBC, THCV and CBGA in human HaCaT cells and used non-cytotoxic half-IC_50_ doses to investigate the possible modulation of the major ECS elements in the same cells. All pCBs were able to modify the gene and protein expression of both eCB-binding receptors and metabolic enzymes to different extents and remarkably, they all also induced the increased protein expression of TRPV1, NAPE-PLD and MAGL and decreased the expression of CB_2_ and PPARδ. In this context, it should be noted that discrepancies between the mRNA and protein expression of specific targets in treated cells are not quite unprecedented and may be due to compensatory changes that affect the half-life of mRNA and the protein turnover. Indeed, similar disparities between changes in mRNA abundance and protein content have been already reported [47], as have those within the ECS by others [48] and by us [34,49]. In addition, it is widely accepted that although the differences in their chemical structures are apparently small, various pCBs may exhibit markedly different biological and pharmacological activities [17]; thus, they can exert different modulations of ECS elements. Our presented functional data showed that CB_1/2_ binding was increased at least twofold by all pCBs that were tested, which extends the previous literature data by showing CBC as a CB_2_ agonist [28,50,51] and CBG and its precursor CBGA as partial agonists of CB_2_ and, to a lesser extent, CB_1_ [52]. Finally, THCV appeared to bind equally well to both CB_1_ and CB_2_ and perform as a CB_1_ antagonist and a partial CB_2_ agonist, both in vivo and in vitro [53]. In general, the ability of rare pCBs to activate CB_2_ may have a major impact on anti-inflammatory and immunomodulatory processes [54,55]. Regarding other eCB-binding receptors, there is growing evidence to suggest that pCBs modulate the activity of GPR55, PPARs and TRPs [56,57], the latter of which are deeply involved in skin sensation, homeostasis and inflammation [58]. Consistent with this evidence, CBG and THCV were shown to have a high effectiveness in human TRPV1 that was over-expressed by HEK-293 cells, as did CBC, albeit to a lesser extent [41]. This previous evidence was fully supported by our data on TRPV1 channel stimulation and protein content.

As well as eCB-binding receptors, the eCB metabolic enzymes were also affected by the rare pCBs, which were all able to increase FAAH and MAGL activity, as well as DAGLα/β (apart from CBGA, which only increased NAPE-PLD). Our analysis of the overall effects of the pCBs on the synthesis of AEA (measured as the NAPE-PLD to FAAH ratio) and 2-AG (the DAGLα/β to MAGL ratio) seemed to suggest that CBG was ineffective, CBC and THCV reduced the former while increasing the latter and CBGA increased the former while reducing the latter (Table 4). However, our analysis of the endogenous content of eCB(-like) compounds showed that CBG was the only substance that was able to increase 2-AG and PEA levels; all of the other pCBs were ineffective. The enzyme activity data and eCB(-like) compounds quantification apparently contrasted with each other, yet it should be recalled that at least 20 additional biosynthetic and hydrolytic enzymes (besides NAPE-PLD, FAAH, DAGLα/β and MAGL) are known to affect the tone of eCBs and eCB-like substances [9,10,11]. In the absence of functional assays, it remains to be ascertained whether pCBs can modulate these enzymes. At any rate, it seems noteworthy that CBG markedly, yet not significantly, increased the DAGLα/β to MAGL activity ratio (Table 4) and doubled CB_1/2_ binding (Table 3), thus suggesting the selective enhancement of 2-AG signaling in human keratinocytes by this pCB. Moreover, the increase in the anti-inflammatory PEA following CBG treatment seems to be of note because a recent study showed that CBG, when applied topically, promotes skin health by reducing the appearance of redness and improving barrier function via anti-inflammatory activity [59].

## 4. Materials and Methods

### 4.1. Materials

The phytocannabinoids (CBG, CBC, THCV and CBGA) were purchased from Cerilliant Corporation (Sigma-Aldrich Company, St. Louis, MO, USA). Cell culture reagents, including high-glucose Dulbecco’s Modified Eagle Medium (DMEM-HG) and fetal bovine serum (FBS), were purchased from Corning Incorporated (Corning, NY, USA). Antibiotics (Pen Strep), phosphate buffer saline (D-PBS without calcium and magnesium) and trypsin (2.5%) were from Gibco by Life Technologies (Thermo Fisher Scientific Company, Waltham, MA, USA) and EDTA (0.5 M) was from Invitrogen (Thermo Fisher Scientific Company, Waltham, MA, USA). The MTT (3-(4,5-dimethylthiazol-2-yl)-2,5-diphenyltetrazolium bromide) reagent was purchased from Sigma-Aldrich (St. Louis, MO, USA). The enzyme-linked immunosorbent assay (ELISA) kit (Cell Death Detection ELISA cod. 11544675001) was purchased from Roche Diagnostic (Basilea, Switzerland). Supplies for the quantitative real-time polymerase chain reaction (RT-qPCR), including the RevertAid H Minus First Strand cDNA Synthesis Kit and SensiFAST SYBR Lo-ROX Kit, were obtained from Thermo Fisher Scientific (Waltham, MA, USA) and Bioline (Meridian Bioscience Inc. Company, Cincinnati, OH, USA), respectively, whereas supplies for the Western blotting were purchased from Bio-Rad Laboratories (Hercules, CA, USA) and the Stripping Buffer was purchased from Thermo Fisher Scientific (Waltham, MA, USA). The UHPLC grade solvents for the UHPLC-MS/MS, including water, methanol, acetonitrile and chloroform, were purchased from VWR International (Radnor, PA, USA), while the deuterated internal standards (AEA-d8, 2-AG-d8 and PEA-d4) and analytical standards (AEA, 2-AG and PEA) were from Vinci-Biochem s.r.l. (Vinci, FI, Italy).

### 4.2. Cell Line and Treatment

Immortalized HaCaT cells from the original depositor (DKFZ, Heidelberg) were purchased from CLS-Cell Lines Service (code 300493) and were of Caucasian skin type (phototype). In all experiments, the HaCaT cells were cultured at 37 °C in a humidified 5% CO_2_ atmosphere in high-glucose Dulbecco’s Modified Eagle Medium (DMEM-HG), which was supplemented with 10% fetal bovine serum, and were used at the 5th–6th passage. Specific pCBs (CBG, CBC, THCV and CBGA) were used to treat the HaCaT cells at different time points (6 h, 12 h and 24 h) and at different concentrations (0.5 µM, 1.0 µM, 2.5 µM, 5.0 µM, 10 µM and 25 µM). Various doses of these pCBs were added directly to culture medium (with 1% fetal bovine serum) and the vehicles alone were added to controls, as follows: 0.8% methanol for CBG and CBC; 0.6% methanol for THCV; 0.6% acetonitrile for CBGA.

### 4.3. MTT Cytotoxicity Assay

The HaCaT cells were seeded into 96-well plates at a density of 1 × 10^4^ per well, incubated overnight and then treated with the selected pCBs at different concentrations (0.5 μM, 1.0 μM, 2.5 μM, 5.0 μM, 10 μM and 25 μM) for 6 h, 12 h and 24 h. Cell viability was assessed using a mitochondrial-dependent reduction of 3-[4,5-dimethylthiazol-2-yl]-2,5-diphenyl tetrazolium bromide (MTT) to purple formazan. After 4 h, the MTT solution was discarded and dimethyl sulfoxide (DMSO) was added to dissolve the formazan crystals. The cell viability was calculated by subtracting the 630 nm OD background from the 570 nm OD total signal of the cell-free blank of each sample and was expressed as percentage of the control (100%).

### 4.4. Determination of Apoptosis

The HaCaT cells were seeded into 96-well plates at a density of 1 × 10^4^ per well, incubated overnight and then treated with the selected pCBs at different concentrations (0.5 μM, 1.0 μM, 2.5 μM, 5.0 μM, 10 μM and 25 μM) for 24 h. After treatment, apoptotic cell death was quantified for specific compounds using ELISA assays, which were based on the evaluation of DNA fragmentation through an immunoassay for histone-associated DNA fragments in the cell cytoplasm. The specific determination of mono- and oligonucleosomes in the cytoplasmic fraction of cell lysates was performed by measuring the absorbance values at 405 nm, as reported in [60].

### 4.5. Cell Cycle Analysis

The HaCaT cells were seeded into 6-well plates at a density of 4 × 10^4^ per well, incubated overnight and then treated with the selected pCBs at three concentrations (1.0 μM, 5.0 μM and 10 μM) for 24 h. After each treatment, the cells were pelleted at 300× *g* for 5 min and 1 × 10^6^ cells were resuspended in 500 μL of ice-cold PBS, which was fixed by a drop-wise addition of an equal volume of ice-cold methanol:acetone (4:1 *v/v*) while vortexing. Upon analysis, the cells were centrifuged, washed in PBS, incubated for 15 min with 13 kunits/mL RNase A from Sigma-Aldrich (St. Louis, MO, USA) and stained with PI (50 μg/mL) from Sigma-Aldrich (St. Louis, MO, USA). The distribution of cells in the G0/G1, S and G2/M phases was determined using an FACS Calibur instrument, following the exclusion of debris and aggregate populations, as well as subG1 populations, by the means of the FlowJo Software, as reported in [61].

### 4.6. Quantitative Real-Time Polymerase Chain Reaction (qRT-PCR)

The HaCaT cells were seeded into 6-well plates at a density of 4 × 10^4^ per well, incubated overnight and then treated with the selected pCBs (CBG (6.0 µM), CBC (4.0 µM), THCV (9.0 µM) and CBGA (13.0 µM)) for 24 h. The cell pellets were collected and RT-PCR reactions were performed using the RevertAid H Minus First Strand cDNA Synthesis Kit. The relative abundance was assessed via RT quantitative PCR (RT-qPCR) using the SensiFAST SYBR Lo-ROX Kit in an Applied Biosystems 7500 Fast Real-Time PCR System (Life Technologies). To provide the precise quantification of the initial target in each PCR reaction, the amplification plot was examined, as well as the point of early log phase of the product accumulation, which was defined by assigning a fluorescence threshold above the background that was defined as the threshold cycle number or “Ct. Differences” in threshold cycle number that was used to quantify the relative amount of the PCR targets that were contained within each tube. After PCR, a dissociation curve (melting curve) was constructed in the range of 60 °C to 95 °C [62] to evaluate the specificity of the amplification products. The relative expression of the different amplicons was calculated using the delta–delta Ct (ΔΔCt) method and then converted into the relative expression ratio (2^−ΔΔCt^) for statistical analysis [63]. All data were normalized to the endogenous reference genes β-actin and glyceraldehyde-3-phosphate dehydrogenase. The specific primers (Table 5) that were used were designed and ordered from Integrated DNA Technologies (IDT; Coralville, IA, USA) with the following sequences:

### 4.7. Western Blotting

The HaCaT cells were seeded into 6-well plates at a density of 4 × 10^4^ per well, incubated overnight and then treated with the selected pCBs (6.0 µM of CBG, 4.0 µM of CBC, 9.0 µM of THCV and 13.0 µM of CBGA) for 24 h. The cell pellets were collected and then lysed in an ice-cold lysis buffer (10 mM of EDTA, 50 mM of pH 7.4 Tris–HCl, 150 mM of sodium chloride, 1% Triton-X-100, 2 mM of phenylmethylsulfonyl fluoride, 2 mM of sodium orthovanadate, 10 mg/mL of leupeptin and 2 mg/mL of aprotinin). The amount of protein was determined using the Bio-Rad Protein assay (Bio-Rad Laboratories, Hemel Hempstead, UK). An equal amount of protein (60 µg) was loaded onto 10% sodium dodecyl sulfate–polyacrylamide gels and blotted onto polyvinylidene fluoride sheets (Amersham Biosciences, Piscataway, NJ, USA). The membranes were blocked with dried 5% non-fat milk for 1 h and then incubated with suitable primary antibodies. Detection was performed using Azure Biosystems c400 (Dublin, CA, USA). The membranes were stripped using the Restore Western Blot Stripping Buffer from Thermo Fisher Scientific (Waltham, MA, USA) for 30 min under agitation at room temperature, according to the manufacturer’s instructions. The primary antibodies (Table 6) that were used were obtained from the following sources:

### 4.8. Receptor Binding Assay on Adherent Living Cells

The HaCaT cells were seeded into 12-well plates at a density of 2 × 10^4^ per well, incubated overnight and then treated with the selected pCBs (6.0 µM of CBG, 4.0 µM of CBC, 9.0 µM of THCV and 13.0 µM CBGA) for 24 h. Each well of the 12-well plates was washed twice with 1 mL of PBS and treated with 500 µL of incubation buffer (50 mM of Tris–HCl, 5 mM of MgCl_2_ and 1 mM of CaCl_2_; 0.2% BSA; pH 7.4), preheated to 37 °C and then incubated for 15 min at 37 °C. Then, 2.5 nM of [^3^H]CP 55,940 was added and incubation occurred for 1 h in an incubator that was set at 37 °C. Afterward, the buffer was carefully removed and the cells were washed again with an ice-cold washing buffer (50 mM of Tris–HCl and 500 mM of NaCl; 0.1% BSA; pH 7.4). Unspecific binding was determined in the presence of 1 µM of synthetic agonist CP 55,940 [60]. Then, 500 µL of 0.5 M NaOH was added to each well and the cells were pipetted up and down several times in order to be lysed. The resuspension was then transferred to a 10-mL scintillation vial with a liquid scintillation cocktail and radioactivity was immediately read using a scintillation β-counter (Tri-Carb 2810 TR, Perkin Elmer, Waltham, MA, USA), as described in [40].

### 4.9. TRPV1 Calcium Assay

The effects of the pCBs on intracellular Ca^2+^ release that was induced by TRPV1 activation was determined using the selective intracellular fluorescent probe Fluo-3 AM (Molecular probes, Eugene, OR, USA). The HaCaT cells (1 × 10^6^ cells) were loaded for 15 min at 25 °C with 4 µM of Fluo-3 AM, which contained 0.02% Pluronic F-127 (Molecular probes) in an OPTIMEM medium, and then they were washed in the Tyrode’s buffer (145 mM of NaCl, 2.5 mM of KCl, 1.5 mM of CaCl_2_, 1.2 mM of MgCl_2_, 10 mM of D-glucose and 10 mM of HEPES; pH 7.4), resuspended in 2 mL of the Tyrode’s buffer and transferred into the quartz cuvette of the LS50B spectrofluorometer (Perkin Elmer, Waltham, MA, USA). Fluorescence was measured at 25 °C (excitation at λ = 488 nm; emission at λ = 516 nm) from the HaCaT cells, which had been pre-incubated with each pCB (6.0 µM of CBG, 4.0 of µM CBC, 9.0 µM of THCV and 13.0 µM of CBGA) and then stimulated with the selective TRPV1 agonist capsaicin (1 µM), as reported in [64]. The TRPV1-mediated intracellular Ca^2+^ elevation was expressed as fluorescence intensity (arbitrary units, AU) per 10^6^ cells.

### 4.10. Enzyme Assays

*NAPE-PLD:* The HaCaT cells were seeded into 6-well plates at a density of 4 × 10^4^ per well, incubated overnight and then treated with the selected pCBs (6.0 µM of CBG, 4.0 µM of CBC, 9.0 µM of THCV and 13.0 µM of CBGA) for 24 h. The cell pellets were collected and NAPE-PLD activity was determined in the homogenates of the cells after 24 h of treatment with each pCB. Then, the homogenates were incubated in a reaction mixture that contained 50 mM of Tris–HCl, 0.05% Triton X-100 pH 8.0 and 10 µM of fluorogenic substrate PED6 (*N*-((6-(2,4-Dinitrophenyl)amino)hexanoyl)-2(4,4-Difluoro-5,7-Dimethyl-4-Bora-3a,4a-Diaza-s-Indacene-3-Pentanoyl)-1-Hexadecanoyl-sn-Glycero-3 Phosphoethanolamine, Triethylammonium Salt). The homogenates were pre-incubated with the specific NAPE-PLD inhibitor ARN19874 at 3.8 mM [65] (Cayman Chemical, Ann Arbor, MI, USA) for 30 min at room temperature in order to fully erase the enzyme activity as a control. Fluorescence values were measured with excitation at λ = 485–488 nm and emission at λ = 530 nm at 37 °C by means of continuity kinetics at 30-sec intervals for 30 min in an Enspire multimode plate reader (Perkin Elmer, Waltham, MA, USA), as reported in [66]. NAPE-PLD activity was expressed as florescence intensity (arbitrary units, AU) per min per mg of protein.

*FAAH Assay:* The HaCaT cells were seeded into 6-well plates at a density of 4 × 10^4^ per well, incubated overnight and then treated with the selected pCBs (6.0 µM of CBG, 4.0 µM of CBC, 9.0 µM of THCV and 13.0 µM of CBGA) for 24 h. The cell pellets were collected and FAAH activity was determined using membranes that were isolated from these cells. Briefly, the membranes were incubated at pH 7.4 with 10 μM of [^3^H]AEA and 50 mM of Tris–HCl, 0.05% BSA at 37 °C in water bath for 30 min. The reaction was blocked with 1 mL of chloroform–methanol (1:1 *v/v*) and centrifuged at 3000 rpm for 10 min at 4 °C. The aqueous (top) layer was collected and transferred into scintillation vials. The amount of [^3^H]ethanolamine that was released was expressed as pmol per min per mg of protein, as reported in [67].

*DAGL α/β Assay:* The HaCaT cells were seeded into 6-well plates at a density of 4 × 10^4^ per well, incubated overnight and then treated with the selected pCBs (6.0 µM of CBG, 4.0 µM of CBC, 9.0 µM of THCV and 13.0 µM of CBGA) for 24 h. The cell pellets were collected and DAGL α/β activity was determined using membranes that were isolated from these cells using the synthetic substrate p-nitrophenyl butyrate (pNPB) (8 mM) in 50 mM of HEPES pH 7.3 at room temperature for 20 min [68]. The DAGLα/β inhibitor KT172 (1 μM), which was purchased from Sigma-Aldrich (St. Louis, MO, USA), was used to fully erase the enzyme activity as a control.

*MAGL Assay:* The HaCaT cells were seeded into 6-well plates at a density of 4 × 10^4^ per well, incubated overnight and then treated with the selected pCBs (6.0 µM of CBG, 4.0 µM of CBC, 9.0 µM of THCV and 13.0 µM of CBGA) for 24 h. The cell pellets were collected and MAGL activity was determined using membranes that were isolated from these cells using the synthetic substrate arachidonoyl-1-thio-glycerol (200 μM) in 10 mM of Tris and 1 mM of EDTA pH 7.2 at 4 °C for 15 min [68]. The MAGL inhibitor JZL184 (4 μM), which was purchased from Sigma-Aldrich (St. Louis, MO, USA), was used to fully erase the enzyme activity as a control.

### 4.11. Quantitation of Endogenous Levels of eCBs and PEA

The HaCaT cells were seeded into 6-well plates at a density of 4 × 10^4^ per well, incubated overnight and then treated with the selected pCBs (6.0 µM of CBG, 4.0 µM of CBC, 9.0 µM of THCV and 13.0 µM of CBGA) for 24 h. The analytical standards that were used were *N*-arachidonoylethanolamine (AEA), *N*-arachidonoylethanolamine-d8 (AEA-d8), 2-arachidonoylglicerol (2-AG), 2-arachidonoylglicerol-d8 (2-AG-d8), *N*-palmitoylethanolamine (PEA) and *N*-palmitoylethanolamine-d8 (PEA-d4). The lipid fraction from these cells was extracted using chloroform–methanol–water (2:1:1 *v/v*) in the presence of ISs (1 ng/mL^−1^ of AEA-d8, 100 ng mL^−1^ of 2-AG-d8 and 1 ng mL^−1^ of PEA-d4). The organic phase was dried under gentle nitrogen stream and then subjected to a micro-solid phase extraction (µSPE) procedure [69] for a rapid clean-up using OMIX C18 tips from Agilent Technologies (Santa Clara, CA, USA). All analyses were performed using a Nexera XR LC 20 AD UHPLC system (Shimadzu Scientific Instruments, USA) that was equipped with Kinetex XB-C18 1.7 µm 100 × 2.1 mm from Phenomenex (Torrance, CA, USA) (for the LC parameters, see Appendix A) and coupled with a 4500 Qtrap from Sciex (Toronto, ON, Canada) that was equipped with a Turbo V electrospray ionization (ESI) source (the MS/MS parameters are shown in Appendix A). The levels of AEA, 2-AG and PEA were then calculated as pmoles per 10^6^ cells, as reported in [60].

## 5. Conclusions

In conclusion, our results demonstrated that CBG, CBC, THCV and CBGA modulate the gene and protein expression of distinct ECS elements differently, as well as their functional activity and their content of eCBs (AEA and 2-AG) and PEA. Notably, all pCBs increased CB_1/2_ binding, TRPV1 channel simulation and FAAH and MAGL activity, which is an observation that represents a proof of concept that they are indeed endowed with biological activity in human keratinocytes via ECS modulation. On this basis, preclinical studies and clinical trials (at least those that are related to skin disorders) should take into account the potential contribution of rare pCBs to the fine-tuning of eCB signaling and biological activity during treatment with cannabis extracts. Of course, these rare pCBs could be used in their pure form as therapeutic drugs, provided that a careful biochemical profiling of their targets and potential off-targets outside the ECS is preliminarily performed. Only then can their therapeutic potential be exploited by ruling out the harmful effects that are often observed cannabis extracts are used as recreational drugs.

## Figures and Tables

**Figure 1 ijms-23-05430-f001:**
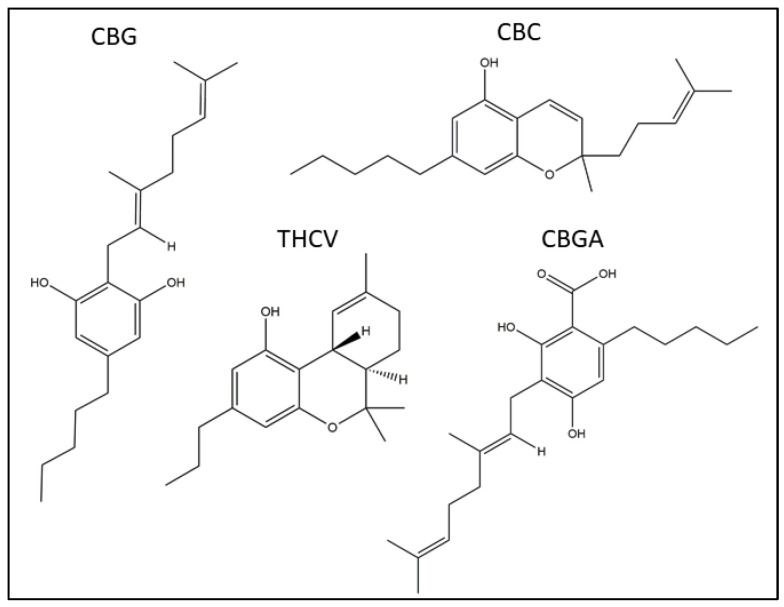
Chemical structures of the four pCBs under study: cannabigerol (CBG), cannabichromene (CBC), Δ^9^-tetrahydrocannabivarin (THCV) and cannabigerolic acid (CBGA).

**Figure 2 ijms-23-05430-f002:**
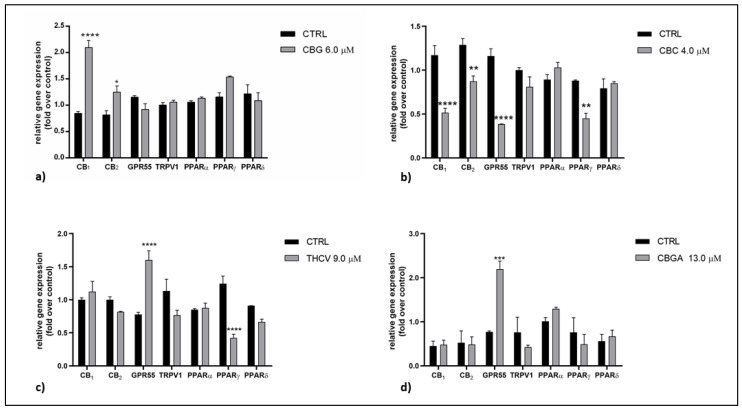
Gene expression of eCB-binding receptors (CB_1_, CB_2_, GPR55, TRPV1 and PPARα/γ/δ) following 24 h of treatment with half-IC_50_ amounts of: (**a**) CBG, (**b**) CBC, (**c**) THCV and (**d**) CBGA. The values were expressed as 2^(−ΔΔCt)^ and were normalized to β-actin and GAPDH as housekeeping genes. The data are the means ± SEM of three independent experiments (*n* = 3). Statistical analysis was performed using a two-way ANOVA test, followed by the Bonferroni post hoc test (* *p* < 0.05, ** *p* < 0.01, *** *p* < 0.001 and **** *p* < 0.0001 vs. control).

**Figure 3 ijms-23-05430-f003:**
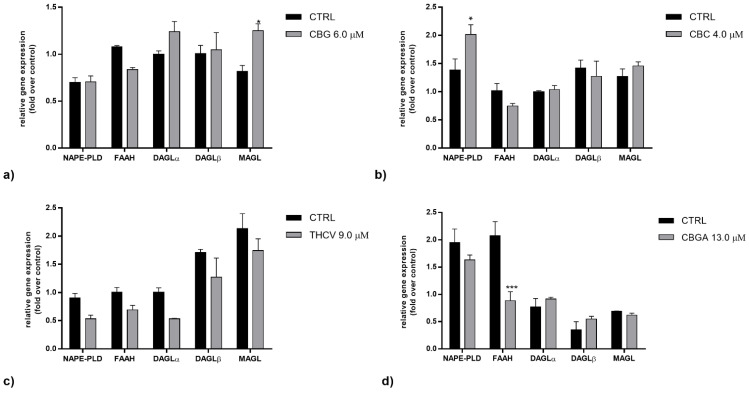
Gene expression of AEA and 2-AG metabolic enzymes (NAPE-PLD, FAAH, DAGLα/β and MAGL) following 24 h of treatment with half-IC_50_ amounts of: (**a**) CBG, (**b**) CBC, (**c**) THCV and (**d**) CBGA. The values were expressed as 2^(−ΔΔCt)^ and were normalized to β-actin and GAPDH as housekeeping genes. The data are the means ± SEM of three independent experiments (*n* = 3). Statistical analysis was performed using a two-way ANOVA test, followed by the Bonferroni post hoc test (* *p* < 0.05 and *** *p* < 0.001 vs. control).

**Figure 4 ijms-23-05430-f004:**
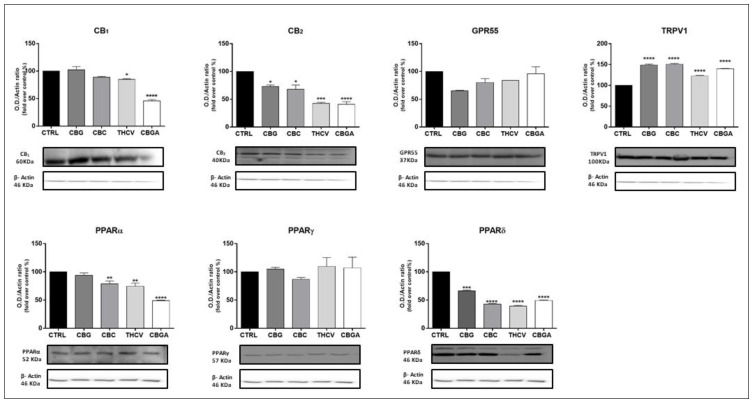
Protein expression of eCB-binding receptors (CB_1_, CB_2_, GPR55, TRPV1 and PPARα/γ/δ) following 24 h of treatment with half-IC_50_ amounts of: CBG, CBC, THCV and CBGA. Densitometric analyses of eCB-binding receptors immunoreactive bands were normalized to β-actin as the housekeeping protein and were presented as fold change over control (%). The data are the means ± SEM of three independent experiments (*n* = 3). Statistical analysis was performed using a one-way ANOVA test, followed by the Bonferroni post hoc test (* *p* < 0.05, ** *p* < 0.01, *** *p* < 0.001 and **** *p* < 0.0001 vs. control).

**Figure 5 ijms-23-05430-f005:**
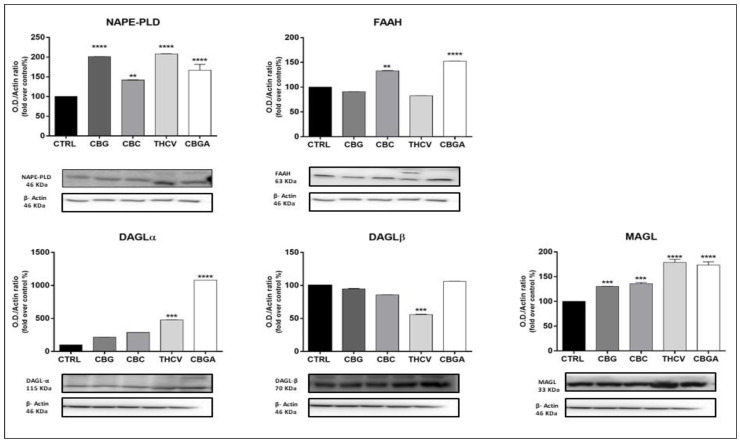
Protein expression of AEA and 2-AG metabolic enzymes (NAPE-PLD, FAAH, DAGLα/β and MAGL) following 24 h of treatment with half-IC_50_ amounts of: CBG, CBC, THCV and CBGA. Densitometric analyses of immunoreactive bands were normalized to β-actin as the housekeeping protein and were presented as fold change over control (%). The data are the means ± SEM of three independent experiments (*n* = 3). Statistical analysis was performed using a one-way ANOVA test, followed by the Bonferroni post hoc test (** *p* < 0.01, *** *p* < 0.001 and **** *p* < 0.0001 vs. control).

**Figure 6 ijms-23-05430-f006:**
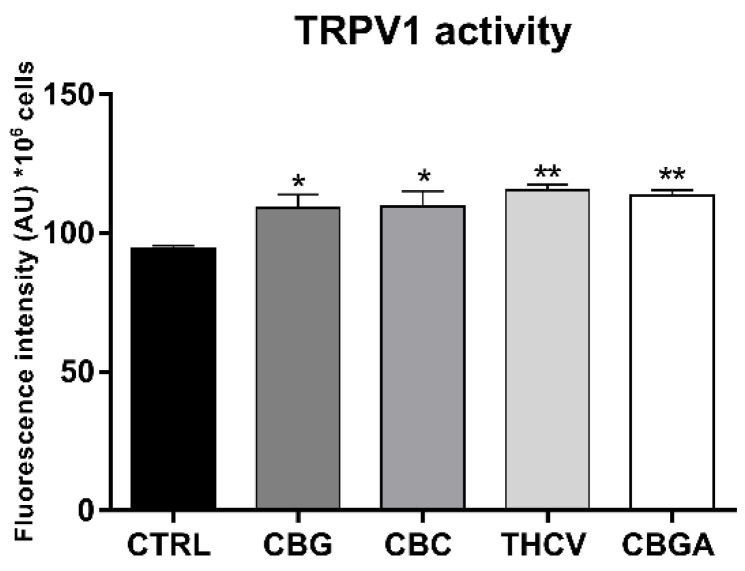
Effects of 24 h of treatment of HaCaT cells with half-IC_50_ amounts of CBG, CBC, THCV and CBGA on intracellular Ca^2+^ release that was triggered by the selective TRPV1 agonist capsaicin. Data are the means ± SEM of three independent experiments (*n* = 3). Statistical analysis was performed using a one-way ANOVA test, followed by the Bonferroni post hoc test (* *p* < 0.05 and ** *p* < 0.01 vs. control).

**Figure 7 ijms-23-05430-f007:**
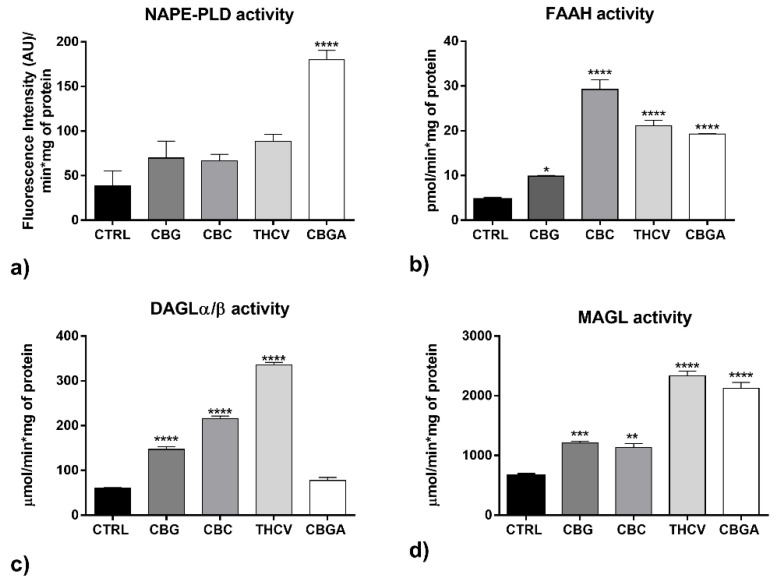
Activity of eCB metabolic enzymes: (**a**) NAPE-PLD, (**b**) FAAH, (**c**) DAGLα/β and (**d**) MAGL following 24 h of treatment with half-IC_50_ amounts of CBG, CBC, THCV and CBGA. Data are the means ± SEM of three independent experiments (*n* = 3). Statistical analysis was performed using a one-way ANOVA test, followed by the Bonferroni post hoc test (* *p* < 0.05, ** *p* < 0.01, *** *p* < 0.001 and **** *p* < 0.0001 vs. control).

**Figure 8 ijms-23-05430-f008:**
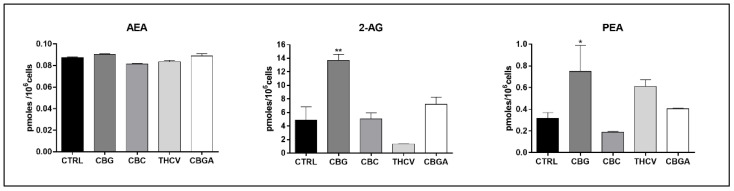
Endogenous levels of eCBs following 24 h of treatment with half-IC_50_ amounts of CBG, CBC, THCV and CBGA. Data are the means ± SEM of two independent experiments (*n* = 2). Statistical analysis was performed using a one-way ANOVA test, followed by the Bonferroni post hoc test (* *p* < 0.05 and ** *p* < 0.01 vs. control).

**Table 1 ijms-23-05430-t001:** IC_50_ values (µM) of CBG, CBC, THCV and CBGA when tested at different time points.

pCB	Time Point	IC_50_ (µM)
CBG	6 h 12 h 24 h	13.5 13.0 11.7
CBC	6 h 12 h 24 h	15.3 15.6 8.2
THCV	6 h 12 h 24 h	14.3 22.4 18.6
CBGA	6 h 12 h 24 h	>25.0 24.5 25.3

**Table 2 ijms-23-05430-t002:** Summary of the most significant effects of half-IC_50_ doses of pCBs on ECS elements at the gene and protein levels following HaCaT cell treatment for 24 h. Legend: ↑ significant increase; ↓ significant decrease; + *p* < 0.05, ++ *p* < 0.01, +++ *p* < 0.001 and ++++ *p* < 0.0001 vs. control.

pCBs	Gene Expression		Protein Expression	
	Receptors	Enzymes	Receptors	Enzymes
CBG	CB_1_ ↑++++ CB_2_ ↑+	MAGL ↑+	CB_2_ ↓+ TRPV1 ↑++++ PPARδ ↓+++	NAPE-PLD ↑++++ MAGL ↑+++
CBC	CB_1_ ↓++++ CB_2_ ↓++ GPR55 ↓++++ PPARγ ↓++	NAPE-PLD ↑+	CB_2_ ↓+ TRPV1 ↑++++ PPARα ↓++ PPARδ ↓++++	NAPE-PLD ↑++ MAGL ↑+++ FAAH ↑++
THCV	GPR55 ↑++++ PPARγ ↓++++		CB_1_ ↓+ CB_2_ ↓+++ TRPV1 ↑++++ PPARα ↓++ PPARδ↓++++	NAPE-PLD ↑++++ DAGLα ↑+++ DAGLβ ↓+++ MAGL ↑++++
CBGA	GPR55 ↑+++	FAAH ↓+++	CB_1_ ↓++++ CB_2_ ↓++++ TRPV1 ↑++++ PPARα ↓++++ PPARδ ↓++++	NAPE-PLD ↑++++ FAAH ↑++++ DAGLα ↑++++ MAGL ↑++++

**Table 3 ijms-23-05430-t003:** CB_1/2_ binding activity of HaCaT cells following 24 h of treatment with half-IC_50_ amounts of the different pCBs. Data are the means ± SEM of three independent experiments (*n* = 3). Statistical analysis was performed using a one-way ANOVA test, followed by the Bonferroni post hoc test (* *p* < 0.05 and ** *p* < 0.01 vs. control).

HaCaT Cell Treatment	CB_1/2_ Binding Activity (pmol/mg of Protein)
Control	4.033 ± 0.516
CBG	8.027 ± 0.681 *
CBC	10.380 ± 0.750 **
THCV	8.187 ± 0.302 *
CBGA	8.373 ± 0.410 *

**Table 4 ijms-23-05430-t004:** Activity ratios of NAPE-PLD/FAAH and DAGLα,β/MAGL with respect to the control (control was set to 1.00). Data are from Figure 7 (* *p* < 0.05 and *** *p* < 0.001 vs. control).

pCB	NAPE-PLD/FAAH Activity Ratio (Mean ± SEM)	DAGLα,β/MAGL Activity Ratio (Mean ± SEM)
Control	1.00 ± 0.00	1.00 ± 0.00
CBG (6.0 µM)	0.97 ± 0.36	1.33 ± 0.13
CBC (4.0 µM)	0.41 ± 0.21	2.09 ± 0.23 ***
THCV (9.0 µM)	0.71 ± 0.30	1.56 ± 0.01 *
CBGA (13.0 µM)	2.51 ± 1.62	0.40 ± 0.06 *

**Table 5 ijms-23-05430-t005:** List of primer sequences used for qRT-PCR analysis.

Gene	Forward Primer Sequence (5′→3′)	Reverse Primer Sequence (5′→3′)
*cnr1*	CCTTTTGCTGCCTAAATCCAC	CCACTGCTCAAACATCTGAC
*cnr2*	TCAACCCTGTCATCTATGCTC	AGTCAGTCCCAACACTCATC
*gpr55*	ATCTACATGATCAACCTGGC	ATGAAGCAGATGGTGAAGACGC
*trpv1*	TCACCTACATCCTCCTGCTC	AAGTTCTTCCAGTGTCTGCC
*pparα*	TGGGAAGGCAGCGTTGATTA	CTGTGTCCTTCCCACTCTCG
*pparγ*	TGATGTCTTGACTCATGGGTGT	CACGGAGCTGATCCCAAAGT
*pparδ*	AGGTTCCCCAAGAGGGAAGA	CAGGAGGAGACAGTTCCAACC
*napepld*	TTGTGAATCCGTGGCCAACATGG	TACTGCCATGGTGAAGCACG
*Faah*	CCCAATGGCTTAAAGGACTG	ATGAACCGCAGACACAAC
*daglα*	AATGGCTATCATCTGGCTGAGC	TTCCGAGGGTGACATTCTTAGC
*daglβ*	GCGCAAAGTAAACGGCAAGA	CTGCAGCTTGGGCTTTTCAT
*Mgll*	ATGCAGAAAGACTACCCTGGGC	TTATTCCGAGAGAGCACGC
*Actb*	TGACCCAGATCATGTTTGAG	TTAATGTCACGCACGATTTCC
*gapdh*	CAGCCTCAAGATCATCAGCA	TGTGGTCATGAGTCCTTCCA

**Table 6 ijms-23-05430-t006:** List of antibodies used for Western blot analysis.

Antibody	Diluition	Brand
CB_1_ receptor rabbit polyclonal	1:200	Cayman Chemical (Ann Arbor, MI, USA)
CB_2_ receptor rabbit polyclonal	1:200	Cayman Chemical (Ann Arbor, MI, USA)
GPR55 receptor rabbit polyclonal	1:200	Cayman Chemical (Ann Arbor, MI, USA)
TRPV1 rabbit polyclonal	1:1000	OriGene (Rockville, MD, USA)
PPAR α rabbit polyclonal	1:1000	Sigma-Aldrich (St. Louis, MO, USA)
PPAR γ rabbit monoclonal	1:1000	Cell Signaling Technology (Danvers, MA, USA)
PPAR δ rabbit polyclonal	1:750	Invitrogen (Waltham, MA, USA)
NAPE-PLD rabbit polyclonal	1:200	Cayman Chemical (Ann Arbor, MI, USA)
FAAH rabbit polyclonal	1:200	Cayman Chemical (Ann Arbor, MI, USA)
DAGL-α rabbit polyclonal	1:1000	Invitrogen (Waltham, MA, USA)
DAGL- β rabbit monoclonal	1:1000	Cell Signaling Technology (Danvers, MA, USA)
MAGL rabbit polyclonal	1:200	Cayman Chemical (Ann Arbor, MI, USA)
β-Actin rabbit monoclonal	1:1000	Cell Signaling Technology (Danvers, MA, USA)

## Data Availability

Not applicable.

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
