# Peer review of "Effects of Rare Phytocannabinoids on the Endocannabinoid System of Human Keratinocytes"

_ijms, 2022, doi:10.3390/ijms23105430_

Round 1
Reviewer 1 Report
Di Meo et al. are presenting the investigation of several phytocannabinoid molecules in this paper. Every phytocannabinoid is tested in a wide-ranging battery of assays designed to characterize each molecule. The legalization of marijuana for medical purposes in more and more countries renders this endeavour timely.
The selected phytocannabinoids are tested in terms of their ability to affect cell viability, and cell cycle; their ability to modify the binding of the receptors that interact with endocannabinoids and affect endocannabinoid synthesising/degrading enzymes. Due to the study design (24h preincubation of cells with the selected compounds then assays with the cell lysates ) it is not the acute pharmacological effects of these phytocannabinoids that are measured. Instead, the measured changes reflect how 24h incubation with phytocannabinoids alters transcription and/or translation of a number of transcripts/proteins. Though qPCR, Westen and functional data don’t always align, interestingly, receptor function (CB1/2 binding and TRPV1 channel function) is increased by all tested compounds.
Nevertheless, I would be a bit cautious with the phrasing „increased TRPV1 activity”. I think what we see is probably not increased TRPV1 activity, at least not in the sense that the individual TRPV1 channel lets through more calcium ions. My impression is that the pCBid-induced increase of TRPV1 channel protein amount in itself (~30-50%) may be enough to explain the increase in calcium signal (not more than 20%).
Looking at the protein analysis data, it is obvious that some bands are very close to the control band, so the staining could not have been done simultaneously. Please mention the stripping conditions in the Methods.
Minor points:
- Fig 3a – please place the asterisk a bit higher for better visibility
- Line 202: erase one of the two dots at the end of the sentence
- Line 299: replace „module” with „modulate”
- Line 317-318: specify what ELISA kit was used
- Lines 431, 432, 446, 462: replace „cells” with „cells’”
- Line 466: replace „full” with „fully”
- Ref 12 is currently in the same paragraph as ref. 11.
Author Response
Reviewer 1:
Di Meo et al. are presenting the investigation of several phytocannabinoid molecules in this paper. Every phytocannabinoid is tested in a wide-ranging battery of assays designed to characterize each molecule. The legalization of marijuana for medical purposes in more and more countries renders this endeavour timely.
Point 1: The selected phytocannabinoids are tested in terms of their ability to affect cell viability, and cell cycle; their ability to modify the binding of the receptors that interact with endocannabinoids and affect endocannabinoid synthesising/degrading enzymes. Due to the study design (24h preincubation of cells with the selected compounds then assays with the cell lysates) it is not the acute pharmacological effects of these phytocannabinoids that are measured. Instead, the measured changes reflect how 24h incubation with phytocannabinoids alters transcription and/or translation of a number of transcripts/proteins. Though qPCR, Westen and functional data don’t always align, interestingly, receptor function (CB1/2 binding and TRPV1 channel function) is increased by all tested compounds.
Response: We agree that it would be interesting to explore also acute pharmacological effects of phytocannabinoids, but of course this can be done only in an independent study. Here, we have performed preliminary experiments to check cell viability at different concentrations and time points (6h-12h-24h), and then we decided to focus only on 24h due to the high number of assays needed to interrogate the endocannabinoid system in a systematic manner. We hope that the Referee can accept our point.
Point 2: Nevertheless, I would be a bit cautious with the phrasing „increased TRPV1 activity”. I think what we see is probably not increased TRPV1 activity, at least not in the sense that the individual TRPV1 channel lets through more calcium ions. My impression is that the pCBid-induced increase of TRPV1 channel protein amount in itself (~30-50%) may be enough to explain the increase in calcium signal (not more than 20%).
Response: We agree with the Referee, and indeed the TRPV1-mediated elevation of calcium following treatment with pCBs may be the simple consequence of increased TRPV1 channel protein. We have revised the manuscript accordingly (page 8, lines 205-208; page 11, line 290).
Point 3: Looking at the protein analysis data, it is obvious that some bands are very close to the control band, so the staining could not have been done simultaneously. Please mention the stripping conditions in the Methods.
Response: Information on stripping conditions has been included in both the materials and the specific section of Western Blotting (page 11, lines 328-329; page 13, lines 406-408).
Minor points:
- Fig 3a – please place the asterisk a bit higher for better visibility
- Line 202: erase one of the two dots at the end of the sentence
- Line 299: replace „module” with „modulate”
- Line 317-318: specify what ELISA kit was used
- Lines 431, 432, 446, 462: replace „cells” with „cells’”
- Line 466: replace „full” with „fully”
- Ref 12 is currently in the same paragraph as ref. 11.
Response: All small errors spotted by the referee have been amended.
Reviewer 2 Report
The manuscript entitled "Effects of rare phytocannabinoids on the endocannabinoid system of human keratinocytes" presents the results of a well-designed high-quality study regarding the expression of different components of the endocannabinoid system at mRNA and protein levels in HaCaT cells.
The results are really surprising and the manuscript is well-written, so its publication in IJMS is warranted.
Here are a few minor observations that require a more specific explanation to avoid some misinterpretations for the readers:
- the housekeeping genes used for normalization in mRNA analysis were b-actin and GAPDH (without mentioning, which gene was normalized to which housekeeping gene) whereas all the proteins analyzed were normalized using b-actin. Could this difference cause a misinterpretation of the results?
- the discrepancy between GPR55 mRNA expression and protein expression should be discussed briefly.
- The parameters of the LC-MS/MS analytical method used for the quantification of endocannabinoids should be provided (briefly in the manuscript, in detail in the supplementary file) - mobile phase composition, retention times, mass transitions, the limit of detection/quantification.
Author Response
Reviewer 2:
The manuscript entitled "Effects of rare phytocannabinoids on the endocannabinoid system of human keratinocytes" presents the results of a well-designed high-quality study regarding the expression of different components of the endocannabinoid system at mRNA and protein levels in HaCaT cells.
The results are really surprising and the manuscript is well-written, so its publication in IJMS is warranted.
Here are a few minor observations that require a more specific explanation to avoid some misinterpretations for the readers:
Point 1: The housekeeping genes used for normalization in mRNA analysis were b-actin and GAPDH (without mentioning, which gene was normalized to which housekeeping gene) whereas all the proteins analyzed were normalized using b-actin. Could this difference cause a misinterpretation of the results?
Response: Each gene was normalized to the mean of the two housekeeping used (β-actin and GAPDH), which both showed a constant trend in all the samples analysed, as well as β-actin as housekeeping protein for Western Blot. This seems to be a largely used, well consolidated procedure that allows accurate quantification of mRNA and protein content, as demonstrated by several studies, ours included (Rapino et al., 2019; Molecules.;24(7):1432).
Point 2: The discrepancy between GPR55 mRNA expression and protein expression should be discussed briefly.
Response: Significant results for GPR55 expression were obtained at the gene level for three of the four phytocannabinoids tested, yet no modifications were found at the protein level. Please note that discrepancies between mRNA and protein expression are not unexpected, and indeed are reported in the literature. More details were included in the revised Discussion (page 10, lines 270-275), to better address this point.
Point 3: The parameters of the LC-MS/MS analytical method used for the quantification of endocannabinoids should be provided (briefly in the manuscript, in detail in the supplementary file) - mobile phase composition, retention times, mass transitions, the limit of detection/quantification.
Response: Additional information, as suggested, has been included in the Materials and Methods (page 15, lines 489-491) of the revised manuscript. Moreover, a new table and a new figure with all LC-MS/MS parameters have been included in the revised Supplementary Materials (page 4, Figure S5 and Table S1).